# Cross-Domain Semi-Supervised Organ Detection

**Nian Li**[1]                                                                                           NIAN.LI@TUM.DE

**Morteza Ghahremani**[1,2]                                                    MORTEZA.GHAHREMANI@TUM.DE

**Bailiang Jian**[1,2]                                                                      BAILIANG.JIAN@TUM.DE

**Pascual Tejero Cervera**[1]                                                    PASCUAL.TEJERO@@TUM.DE

**Benedikt Wiestler**[1,2]                                                                   B.WIESTLER@TUM.DE

**Marcus Makowski**[1]                                                             MARCUS.MAKOWSKI@TUM.DE

**Christian Wachinger**[1,2]                                               CHRISTIAN.WACHINGER@TUM.DE

[1] *Technical University of Munich (TUM),* [2] *Munich Center for Machine Learning (MCML)*

**Editors:** Accepted for publication at MIDL 2026

## Abstract

Domain adaptation for 3D organ detection in CT imaging is challenging due to variations in scanner types, imaging protocols, and overall acquisition conditions. As supervised detection models require large, annotated datasets from diverse scanners and institutions, semi-supervised approaches have gained attention for their ability to leverage limited unlabeled target data. However, traditional semi-supervised methods typically fail to make effective use of the few labeled target samples and most often do not yield satisfactory results. To address this limitation, we introduce a novel cross-domain semi-supervised detection framework (CDSS-Det) built upon the Transformer-based Organ-DETR model. CDSS-Det is a cross-domain semi-supervised framework for 3D organ detection that addresses unreliable pseudo-labels and limited target supervision under domain shift. It introduces a curriculum-guided pseudo-labeling mechanism and domain-robust representation learning to enable effective knowledge transfer from a well-annotated source domain to a sparsely labeled target domain. Experiments on multi-domain CT datasets demonstrate that incorporating a small number of labeled target samples significantly boosts detection performance over conventional domain adaptation and semi-supervised methods. CDSS-Det consistently achieves higher mean Average Precision (mAP), with notable improvements in detecting small organs, and surpasses a fully supervised model trained solely on the labeled target domain by over 10%. These results underscore the potential of CDSS-Det in efficiently leveraging both labeled and unlabeled target data in cross-domain organ detection, advancing annotation-efficient deep learning models in medical imaging.

**Keywords:** Organ Detection, Domain Transfer, Domain Adaptation

## 1. Introduction

Accurate 3D organ detection from CT scans is essential for disease diagnosis, surgical planning, and downstream applications such as segmentation (Ma et al., 2021). Although deep learning-based object detection models have achieved impressive results on well-annotated datasets (Ghahremani et al., 2025), their generalization to new domains is hindered by substantial domain shifts, arising from variations in scanner types, imaging protocols, and patient demographics. As illustrated in Figure 1, this shift is particularly pronounced in medical imaging, where transferring a model between datasets yields a drastic drop in mean

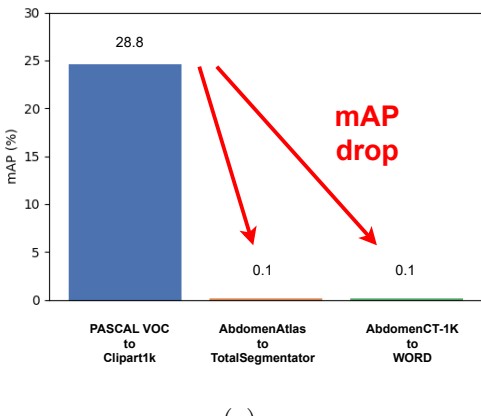
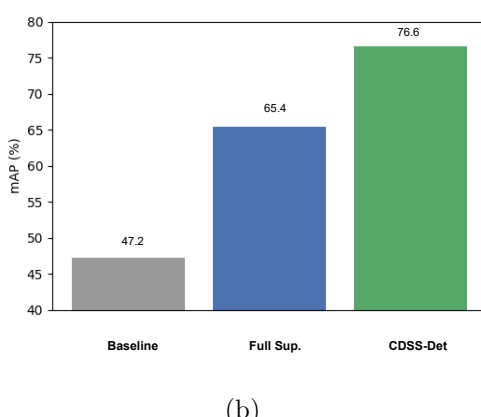

Figure 1: (a) the mAP result of applying a source-trained model directly to a target domain, highlighting the severe domain gap in medical imaging compared to natural image datasets. (b) CDSS-Det improves cross-domain organ detection on the WORD dataset, outperforming both baseline and fully supervised models.

Average Precision (mAP). In contrast, object detection generalizes well on natural image datasets (Li et al., 2022).

Furthermore, developing high-performance object detectors requires large-scale labeled datasets, the annotation of which is both resource-intensive and time-consuming. To overcome this challenge, *domain adaptation* has emerged as a promising approach, enabling models trained on a labeled source domain to effectively generalize to a related but distinct target domain (Guan et al., 2021). Despite progress in domain adaptation, most prior works focus on unsupervised domain adaptation (UDA) (Tzeng et al., 2017; Chen et al., 2020). Although no labeled target data is required in UDA methods, they often struggle in medical imaging due to substantial domain shifts caused by variations in image acquisition and patient demographics (Zhang et al., 2020). *Semi-supervised learning* (SSL) has been explored to address the scarcity of labeled data by leveraging unlabeled target data alongside limited labeled samples (Ouali et al., 2021; Bai et al., 2017). However, existing semi-supervised object detection methods in medical imaging often rely solely on labeled source data or unlabeled target data, making it challenging to achieve satisfactory performance due to domain shifts and the lack of direct supervision on the target domain (Jeong et al., 2019; Sohn et al., 2020). Few-shot learning approaches attempt to mitigate this issue by training on a small set of labeled target samples (Wang et al., 2020), but they fail to utilize the large pool of available unlabeled target data, limiting their ability to generalize effectively.

Recent studies have explored domain adaptation and semi-supervised learning for medical image analysis, primarily in classification and segmentation tasks. For instance, Yuan et al. (Yuan et al., 2024) utilized pseudo-labeling for COVID-19 detection, demonstrating that incorporating unlabeled target data can enhance domain transfer in medical classification. In the segmentation domain, Cai et al. (Cai et al., 2024) proposed a Class-Aware Mutual Mixup strategy with triple alignments, while Basak and Yin (Basak and Yin, 2023) introduced a consistency-regularized disentangled contrastive learning approach, both showing the effectiveness of combining alignment and consistency in pixel-level tasks. Similarly, contrastive learning methods such as PixPro (Xie et al., 2021a) aim to improve feature consistency across domains in self-supervised settings. While these approaches have shown

promising results, they are primarily designed for image-level or pixel-wise prediction and do not directly address the challenges of 3D object detection, which requires accurate instance-level localization and handling of class imbalance under domain shift. Moreover, existing methods are not designed for a practically important setting in medical imaging, where a small number of labeled target scans is available alongside abundant unlabeled target data. This mismatch limits their applicability to real-world cross-domain 3D detection scenarios, where effectively leveraging both limited labeled and abundant unlabeled target data is critical.

To address this gap, we propose cross-domain semi-supervised organ detection (CDSS-Det), a framework specifically designed for this practical setting. CDSS-Det builds upon the Transformer-based Organ-DETR model (Ghahremani et al., 2025) and introduces a curriculum-controlled pseudo-labeling mechanism tailored for cross-domain 3D medical detection. By explicitly exploiting both limited labeled target data and abundant unlabeled target data, the proposed framework enables stable and effective adaptation under large domain shifts. Experimental results demonstrate that CDSS-Det achieves superior mean Average Precision compared to existing baselines (Figure 1), highlighting the effectiveness of the proposed learning strategy for cross-domain organ detection. Our contributions are summarized below and the source code is publicly available at: https://github.com/ai-med/CDSS-Det.

- We define a practical cross-domain semi-supervised setting for 3D organ detection, where labeled source data, limited labeled target data, and abundant unlabeled target data are jointly utilized. This setting better reflects real-world medical imaging scenarios and is underexplored in existing literature.

- We propose a curriculum-guided pseudo-labeling mechanism that dynamically regulates the contribution of pseudo-label supervision based on model confidence, enabling stable and effective learning under domain shift.

- We develop CDSS-Det, a unified framework that incorporates reliability-aware pseudo-label learning and domain-robust representation learning for cross-domain 3D detection. Extensive experiments on two benchmarks demonstrate consistent improvements over strong baselines, with particularly significant gains on small organs.

## 2. Methodology

**Preliminaries**. Organ detection in 3D CT imaging involves localizing anatomical structures using axis-aligned bounding boxes and assigning class labels to detected organs (Shin et al., 2016). Detection performance is evaluated using mAP at different IoU thresholds, mean Average Recall (mAR), precision, and recall. We build upon Organ-DETR (Ghahremani et al., 2025), a Transformer-based 3D object detector designed for medical imaging. It introduces MultiScale Attention (MSA) for handling varying organ sizes and Dense Query Matching (DQM) to improve query-object associations, enhancing detection robustness in CT scans.

**Problem Definition**. We address *cross-domain semi-supervised* organ detection, where a model is trained using a labeled source dataset $D_s = \{(X_i^s, Y_i^s)\}$, a small set of labeled

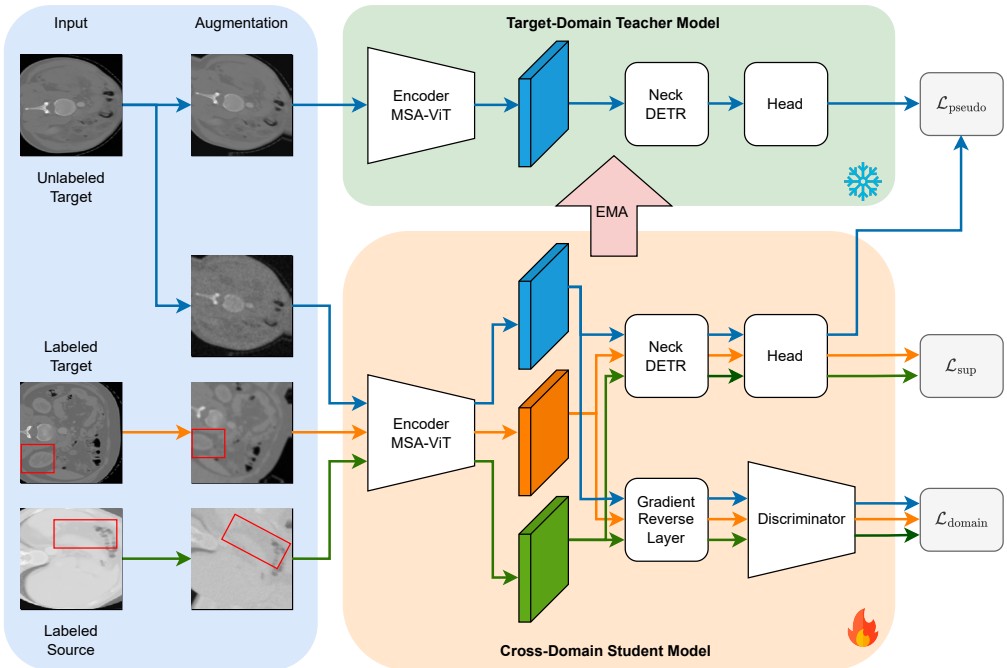

Figure 2: Overview of the CDSS-Det framework. Both student and teacher branches are based on Organ-DETR, which incorporates Multi-Scale Attention (MSA) and Dense Query Matching (DQM) to enhance 3D organ detection. The teacher model processes unlabeled target data to generate pseudo-labels, while the student model is trained using labeled source data, labeled target data, and unlabeled target data. Supervised loss is computed from labeled predictions, domain loss is obtained via a discriminator with a gradient reversal layer, and pseudo loss is calculated between student predictions and pseudo labels. The teacher is updated via Exponential Moving Average (EMA) of the student weights.

target samples $D_t = \{(X_i^t, Y_i^t)\}$, and a larger set of unlabeled target scans $U_t = \{X_j^t\}$. This setting reflects practical medical imaging scenarios, where limited annotations are available in the target domain despite substantial domain shifts from the source domain. The primary challenge is the domain gap between $D_s$ and $D_t$, which can significantly degrade detection performance when models trained on $D_s$ are directly applied to $D_t$, as illustrated in Figure 1. While semi-supervised learning provides a natural approach by leveraging unlabeled target data through pseudo-labeling and teacher–student learning (Ouali et al., 2021), existing methods are not designed for this cross-domain setting with limited labeled target data, and often suffer from unreliable pseudo-labels and unstable adaptation under domain shift.

## 2.1. CDSS-Det Framework for Cross-Domain Semi-Supervised Learning

Figure 2 provides an overview of the CDSS-Det framework. We consider a cross-domain semi-supervised setting for 3D organ detection, where labeled source data, a small amount of labeled target data, and abundant unlabeled target data are jointly available. To effectively leverage these heterogeneous data sources, we adopt a teacher–student framework, in which

the teacher model generates pseudo-labels on unlabeled target data, and the student model is trained using both labeled and pseudo-labeled supervision.

However, directly applying standard teacher–student learning in cross-domain medical imaging is challenging due to substantial domain shifts across datasets, which can lead to unreliable pseudo-labels and unstable training. At the same time, the limited availability of labeled target data restricts the model's ability to adapt to target-specific characteristics, while naive adaptation may degrade the knowledge learned from the source domain.

To address these challenges, CDSS-Det is designed around three key principles: (1) reliability-aware pseudo-label learning to mitigate noise introduced by domain shift, (2) curriculum-guided supervision balancing to regulate the contribution of pseudo-labels based on model confidence, and (3) domain-robust representation learning to align feature distributions while preserving discriminative knowledge. These components are integrated within a unified teacher–student framework to enable stable and effective cross-domain adaptation for 3D organ detection.

**Reliability-Aware Pseudo-Label Learning**: To exploit unlabeled target data while mitigating noise introduced by domain shift, pseudo-labels are filtered and refined based on prediction confidence. A pseudo-label is retained only if its classification confidence exceeds a threshold $\tau$:

$$\hat{y}_i^t = \arg\max p(y_i|X_i^t), \quad \text{if} \quad p(y_i|X_i^t) > \tau, \tag{1}$$

where $X_i^t$ is the $i$-th unlabeled target sample, $y_i$ represents the ground-truth class label, and $\hat{y}_i^t$ is the assigned pseudo-label. To eliminate redundant detections, we further apply IoU-based Non-Maximum Suppression (NMS).

Due to domain shift, bounding box predictions from pseudo-labels may remain unreliable. Therefore, instead of applying regression losses, we use classification-only supervision for pseudo-labels. The pseudo-label loss is defined as:

$$\mathcal{L}_{\text{pseudo}} = \frac{1}{N_p} \sum_{i=1}^{N_p} \mathcal{L}_{\text{CE}}(y_i^t, \hat{y}_i^t). \tag{2}$$

**Curriculum-Guided Supervision Balancing**: In cross-domain settings, pseudo-labels are inherently less reliable than labeled data due to domain shift, and directly applying them with fixed weighting can lead to confirmation bias and unstable optimization. This issue is particularly pronounced in 3D medical detection, where small anatomical variations can significantly affect prediction confidence.

To address this, we design a curriculum mechanism that dynamically regulates the contribution of pseudo-label supervision based on the model's confidence on labeled data. Instead of treating pseudo-labels as equally reliable throughout training, the model gradually increases their influence only when it demonstrates sufficient confidence on labeled samples. Formally, the pseudo-label weight is updated as:

$$\lambda_{\text{pseudo}}(t) = \begin{cases} \min(\lambda_{\text{pseudo}}(t-1) + \Delta, \lambda_{\max}), & \text{if } \mathcal{L}_{\text{cls}} < \delta \\ \max(\lambda_{\text{pseudo}}(t-1) - \Delta, \lambda_{\min}), & \text{if } \mathcal{L}_{\text{cls}} \geq \delta. \end{cases} \tag{3}$$

If the student classification loss on labeled data falls below a threshold $\delta$, the pseudo-label weight is increased by $\Delta$, up to $\lambda_{\max}$. Otherwise, it is decreased but bounded by $\lambda_{\min}$.

This design introduces a feedback-driven curriculum that adapts to the learning state of the model, ensuring that pseudo-label supervision is introduced progressively and remains

bounded. As a result, the model avoids over-reliance on noisy pseudo-labels in early stages while fully exploiting unlabeled data once reliable representations are learned. This mechanism is particularly effective in scenarios with limited labeled target data, where balancing supervised and pseudo-supervised signals is critical for stable adaptation.

**Domain-Robust Representation Learning**: To address both domain discrepancy and the risk of forgetting source knowledge, we combine adversarial alignment with a replay strategy. A domain discriminator is attached to backbone features and trained to distinguish between source and target domains, while the student model learns domain-invariant representations through a gradient reversal layer (GRL) (Zhang et al., 2020).

At the same time, we incorporate a replay mechanism inspired by continual learning (Rolnick et al., 2019). Instead of randomly sampling source data, we identify and replay the hardest labeled source samples based on detection confidence. These informative samples are jointly trained with target data, reinforcing discriminative features and stabilizing adaptation. The combination of adversarial alignment and targeted replay enables more robust feature learning under domain shift.

**Teacher–Student Optimization**: The teacher–student framework is optimized jointly using supervised, pseudo-label, and domain alignment objectives. The teacher model is maintained as an Exponential Moving Average (EMA) of the student model parameters:

$$\theta_t \leftarrow \alpha\theta_t + (1-\alpha)\theta_s, \tag{4}$$

where $\theta_t$ and $\theta_s$ denote the teacher and student parameters, respectively, and $\alpha$ is the EMA decay factor. This design stabilizes pseudo-label generation and provides a consistent training signal.

The overall training objective for the student integrates supervised learning on labeled data, pseudo-label learning on unlabeled data, and domain alignment:

$$\mathcal{L}_{\text{student}} = \mathcal{L}_{\text{sup}} + \lambda_{\text{pseudo}}(t)\mathcal{L}_{\text{pseudo}} + \lambda_{\text{domain}}\mathcal{L}_{\text{domain}}. \tag{5}$$

The supervised loss is computed on both labeled source and labeled target data:

$$\mathcal{L}_{\text{sup}} = \mathcal{L}_{\text{sup}}^{\text{src}} + \mathcal{L}_{\text{sup}}^{\text{tgt}}, \tag{6}$$

where each term consists of classification, localization, and segmentation components:

$$\mathcal{L}_{\text{sup}}^{(\cdot)} = \mathcal{L}_{\text{cls}}^{(\cdot)} + \mathcal{L}_{\text{bbox}}^{(\cdot)} + \mathcal{L}_{\text{giou}}^{(\cdot)} + \mathcal{L}_{\text{seg}}^{(\cdot)}. \tag{7}$$

Here, $\mathcal{L}_{\text{cls}}^{(\cdot)}$ denotes classification loss, $\mathcal{L}_{\text{bbox}}^{(\cdot)}$ is L1 bounding box regression loss, $\mathcal{L}_{\text{giou}}^{(\cdot)}$ is Generalized IoU loss, and $\mathcal{L}_{\text{seg}}^{(\cdot)}$ is an optional segmentation loss composed of cross-entropy and Dice losses. Segmentation supervision is applied only to labeled data, following the original Organ-DETR design, and serves as an auxiliary signal to enhance multi-scale feature learning. It is not used for unlabeled data, pseudo-label generation, or inference.

$$\mathcal{L}_{\text{seg}}^{(\cdot)} = \mathcal{L}_{\text{ce}}^{(\cdot)} + \mathcal{L}_{\text{dice}}^{(\cdot)}. \tag{8}$$

The curriculum-controlled weight $\lambda_{\text{pseudo}}(t)$ regulates pseudo-label influence based on model confidence, while $\lambda_{\text{domain}}$ scales the domain alignment objective. Together, these components enable stable and effective integration of labeled and unlabeled data for cross-domain 3D organ detection.

## 3. Experiments

**Datasets**.We evaluate our method on two cross-domain 3D organ detection settings. The first setting, AbdomenAtlas → TotalSegmentator, uses AbdomenAtlas, a large-scale, multi-center dataset with annotations for multiple abdominal organs (Qu et al., 2023). We use AbdomenAtlas 1.0 with 5,195 scans, where 3,524 scans were used as the training set to pre-train our model. To the best of our knowledge, this is the first study utilizing AbdomenAtlas for cross-domain, semi-supervised organ detection. The scans in this dataset include both healthy and diseased organs such as tumors and fatty liver. Axis-aligned bounding boxes are extracted from segmentation maps and used as detection labels. Scans are normalized using the 0.5 and 99.5 percentiles of non-background voxels, clipped to the [0, 1] range. Augmentations (applied with 50% probability) include random intensity scaling/shifting (up to 10%), rotation ($\pm 5°$), translation (up to 10%), and zooming ($\pm 10\%$).

In the target dataset, TotalSegmentator (Wasserthal et al., 2022), the training set contains 113 scans, which are split into 8 labeled target scans and 105 unlabeled target scans for semi-supervised training. In addition, 21 scans are used for validation and 29 scans for testing, which are not included in the training set. Eight common organs from these two datasets are selected for our detection task. The TotalSegmentator dataset also includes both healthy and pathological cases.

The second setting, AbdomenCT-1K → WORD, involves AbdomenCT-1K, a dataset of 1,112 high-resolution 3D CT scans from five sources, covering the liver, left kidney, right kidney, spleen, and pancreas (Ma et al., 2021). These scans exhibit variability in slice thickness and pixel spacing, making them suitable for cross-domain adaptation. 732 samples in the training set are used to pre-train the model. The scans contain both healthy and diseased organs, including cancer and tumors. The same normalization and augmentation techniques as above are applied to improve robustness.

The target dataset, WORD, contains 150 CT scans acquired from a single medical center with high-resolution imaging and multiple organ annotations (Miao et al., 2021). We use 31 labeled target scans and 75 unlabeled target scans for training, with 14 validation scans and 29 test scans. Five common organs from these two datasets are selected for our detection task. Similar to other datasets, bounding boxes are derived from segmentation masks.

An overview of dataset splits and organ size definitions is provided in Table 1.

**Training and Evaluation Setup**. Training uses AdamW (weight decay $1 \times 10^{-4}$) with an initial learning rate of $2 \times 10^{-4}$, decaying by 0.1 every 500 epochs, for a total of 2,500 epochs. Each iteration processes one labeled source, one labeled target, and one unlabeled target sample. The supervised loss includes classification, bounding-box regression, and segmentation, with weights: cls = 2, bbox = 5, giou = 2, segce = 2, segdice = 2. Pseudo-labels are filtered with a confidence threshold of 0.8 and refined via NMS (IoU = 0.5). The teacher model is updated using EMA (decay = 0.9996). The pseudo-label weight $\lambda_{\text{pseudo}}(t)$ adjusts by $\pm 0.1$ based on a classification-loss threshold of 0.01, and is clipped to $[0, 2]$. These parameter choices follow general design principles for semi-supervised learning in 3D medical CT detection, where pseudo-label supervision should be bounded relative to labeled supervision and should not dominate it. We validate this design across two substantially different cross-domain settings: two large-scale source datasets used for pre-training (AbdomenAtlas and AbdomenCT-1K), and two target domains with markedly

Table 1: Dataset splits and organ size definitions for the two cross-domain settings. "Labeled" and "Unlabeled" refer to training data only.

| Dataset | Labeled Train | Unlabeled Train | Val | Test | Organ size definitions |
|---------|---------------|-----------------|-----|------|------------------------|
| TotalSeg | 8 | 105 | 21 | 29 | **Large**: gallbladder, pancreas
**Medium**: left kidney, right kidney, spleen, aorta
**Small**: stomach, liver |
| WORD | 31 | 75 | 14 | 29 | **Large**: pancreas
**Medium**: left kidney, right kidney, spleen
**Small**: liver |

different levels of labeled supervision (31 labeled scans in WORD versus only 8 labeled scans in TotalSegmentator). The consistent effectiveness across these settings supports the robustness and general applicability of the proposed pseudo-label confidence threshold and curriculum strategy, rather than sensitivity to precise parameter values. Domain adaptation loss is added with weight 0.2. A replay strategy selects hard source samples with the lowest detection performance (measured by mAP) from the source-trained model, and replays as many of these samples as labeled target samples during training. All experiments are conducted on an NVIDIA A100 GPU (80 GB).

To evaluate CDSS-Det, we compare it against multiple baselines, including a baseline model trained solely on labeled target data, a pre-trained variant initialized with a source-trained model, and a fully supervised (Full Sup.) model trained with full target annotations. Additionally, we conduct an ablation study to assess the contribution of different components within CDSS-Det.

Recent cross-domain semi-supervised methods in classification (Yuan et al., 2024) and segmentation (Cai et al., 2024; Basak and Yin, 2023) have shown encouraging results, but they lack publicly available source code and implementation details, making direct comparisons infeasible. Furthermore, methods such as (Basak and Yin, 2023), which focus on 2D segmentation, are difficult to adapt to 3D object detection due to differences in task formulation and model design. As an alternative, we include PixPro (Xie et al., 2021b) as an additional consistency constraint, implemented as a separate loss to encourage feature consistency. We empirically determine its optimal coefficient to be 0.01 in our setting. Note that all reported results are obtained using the student model during inference.

**Experimental Results**. Table 2 summarizes the detection performance of different training strategies on the WORD and TotalSegmentator datasets. CDSS-Det consistently achieves the highest detection performance across different IoU thresholds and organ sizes, demonstrating its effectiveness in leveraging labeled source, labeled target, and unlabeled target data for cross-domain semi-supervised organ detection. The baseline model, trained only on labeled target data, achieves the lowest performance on both datasets, highlighting the challenges of learning from limited labeled data in the target domain. Pre-training on the

Table 2: Detection performance on the WORD and TotalSegmentator datasets under different training strategies. The baseline is trained solely on labeled target data. The Pre-trained model is initialized with a source-trained model to improve generalization. The Full Sup. assumes access to all target data with full annotations. We report mAP grouped by organ size for small (S), medium (M), and large (L) organs.

| Dataset | Method | mAP ↑ | | | mAR ↑ | | | mAP ↑ by size | | |
|---|---|---|---|---|---|---|---|---|---|---|
| | | Total | 75% | 50% | Total | 75% | 50% | S | M | L |
| WORD | Baseline | 47.2 | 44.7 | 96.5 | 54.0 | 57.9 | 97.2 | 26.2 | 50.4 | 58.7 |
| | Pre-trained | 72.9 | 85.6 | 97.5 | 77.7 | 89.0 | **98.6** | 54.0 | 77.2 | 79.0 |
| | Full Sup. | 65.4 | 78.1 | **98.1** | 71.1 | 87.1 | **98.6** | 40.4 | 71.4 | 72.6 |
| | CDSS-Det | **76.6** | **88.8** | 97.4 | **79.9** | **91.7** | 97.9 | **58.4** | **80.7** | **82.7** |
| TotalSeg | Baseline | 13.7 | 2.1 | 50.7 | 21.0 | 8.2 | 62.5 | 6.0 | 13.9 | 21.0 |
| | Pre-trained | 62.5 | 68.1 | **94.5** | 68.2 | 75.5 | 95.9 | 32.1 | 76.6 | 64.7 |
| | Full Sup. | 52.7 | 62.7 | 88.6 | 59.2 | 69.9 | 91.8 | 20.5 | 64.4 | 61.6 |
| | CDSS-Det | **70.1** | **82.2** | 94.1 | **75.2** | **85.8** | **96.3** | **42.2** | **81.7** | **74.5** |

source dataset significantly improves performance, confirming the importance of transferring knowledge from a larger labeled dataset. Remarkably, the Full Sup. model, despite full supervision, falls short of CDSS-Det, indicating that refined pseudo-labeling can effectively supplement sparse annotations and improve generalization. Figure 3 presents a qualitative comparison of organ detection results between Full Sup. and CDSS-Det.

In clinical practice, 3D organ detection is often used as a localization or initialization step, such as region-of-interest cropping or as a precursor to downstream segmentation, where an IoU threshold around 0.5 is typically sufficient. At the same time, higher IoU thresholds (e.g., 0.75) are important for robust automation, reducing downstream correction effort, and accurately localizing small or anatomically variable organs. As shown in Table 2, CDSS-Det consistently improves performance across IoU thresholds, achieving gains not only at 75% IoU but also at 50% IoU. These improvements are particularly pronounced for small and medium organs, which are most sensitive to localization errors under domain shift, while performance on large organs is preserved.

CDSS-Det provides notable gains, particularly for small organs, which are traditionally difficult to detect due to their anatomical variability and limited representation in training data. As shown in Figure 4, CDSS-Det significantly outperforms the Full Sup. model across all organ sizes, with the largest improvements observed on small organs in both WORD and TotalSegmentator datasets. These results demonstrate that pseudo-label refinement and dynamic weighting are especially effective in addressing the challenges of detecting underrepresented structures.

We note that the relative clinical importance of large, medium, and small organs can vary across applications. Importantly, CDSS-Det does not trade off large-organ performance to achieve gains on smaller organs: performance on large organs remains comparable to or better than competing methods across both datasets. This indicates improved robustness rather than a bias toward specific organ sizes. Moreover, the proposed framework is flexible,

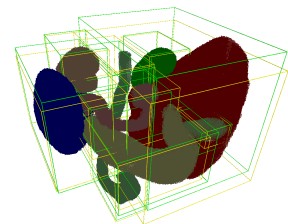
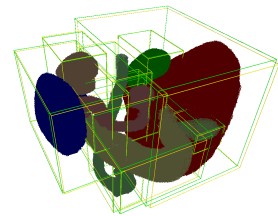

Figure 3: Visualization of organ detection results on WORD dataset. The left shows results from Full Sup. and the right presents results from CDSS-Det. Ground truth bounding boxes are in green, and predicted bounding boxes are in yellow.

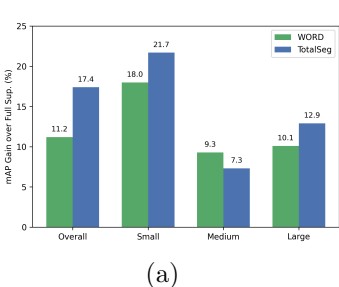
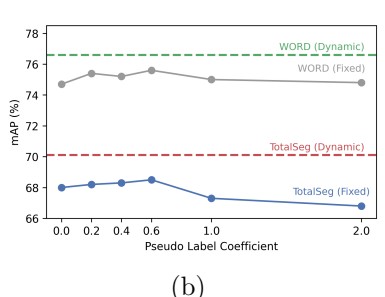
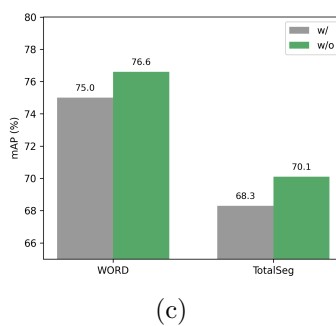

(a)  (b)  (c)

Figure 4: (a) mAP gain of CDSS-Det over the Full Supervised model across different organ sizes (Small, Medium, Large) on the WORD and TotalSegmentator datasets. (b) Ablation study on the pseudo-label loss coefficient strategies, and (c) Ablation study on the weak-strong augmentation impact.

as loss weighting or evaluation emphasis can be adjusted to align with task-specific clinical priorities when required.

Recent works in medical domain adaptation and semi-supervised learning, such as Yuan et al. (Yuan et al., 2024), Cai et al. (Cai et al., 2024), and Basak et al. (Basak and Yin, 2023), have explored related ideas in the context of classification and segmentation. However, these methods focus on image-level classification (Yuan et al., 2024) or pixel-wise segmentation (Cai et al., 2024; Basak and Yin, 2023), and do not address the instance-level challenges inherent in 3D object detection. In addition, they lack open-source implementations and sufficient details for reproducibility, and methods designed for 2D segmentation tasks are not straightforward to adapt to volumetric 3D detection problems. As a result, we include PixPro (Xie et al., 2021a) as a representative self-supervised learning baseline in our ablation study to evaluate the potential of pixel-level consistency in 3D detection tasks.

Table 3 reports results for the ablation study that evaluated the contribution of individual components within CDSS-Det. The replay strategy improves feature stability by retaining harder samples from the source domain. Domain adaptation provides an additional performance boost by reducing feature discrepancies between source and target distributions, but its impact remains relatively small compared to pseudo-labeling. Self-supervised feature consistency with PixPro does not yield substantial improvements, indicating that contrastive learning may be less effective for volumetric medical imaging. Overall, the highest gains come from pseudo-labeling and dynamic weighting, which allow the model to

Table 3: Ablation study on the WORD and TotalSegmentator datasets. Configuration settings include Replay (R), Domain Alignment (D), PixPro (P), Pseudo-Labeling (PL), and Dynamic Pseudo-Labeling (Dyn).

| | Configuration | | | | | mAP↑ | | | mAR↑ | | | mAP ↑ by size | | |
|---|---|---|---|---|---|---|---|---|---|---|---|---|---|---|
| | R | D | P | PL | Dyn | Total | 75% | 50% | Total | 75% | 50% | S | M | L |
| **WORD** | ✓ | | | | | 74.5 | 87.7 | 96.1 | 79.7 | **92.4** | 97.9 | 52.4 | 78.7 | **84.2** |
| | ✓ | ✓ | | | | 74.7 | 86.2 | **97.5** | 79.6 | 91.0 | **98.6** | 52.7 | 79.6 | 82.0 |
| | ✓ | ✓ | ✓ | | | 74.8 | 87.7 | 96.4 | 79.5 | 91.7 | 97.9 | 55.3 | 79.5 | 80.2 |
| | ✓ | ✓ | | ✓ | | 75.6 | 88.2 | 97.4 | 79.5 | 91.7 | 97.9 | 56.6 | 80.0 | 81.3 |
| | ✓ | ✓ | | ✓ | ✓ | **76.6** | **88.8** | 97.4 | **79.9** | 91.7 | 97.9 | **58.4** | **80.7** | 82.7 |
| **TotalSeg** | ✓ | | | | | 67.4 | 78.8 | 92.7 | 73.2 | 84.3 | 95.1 | 40.3 | 78.5 | 72.3 |
| | ✓ | ✓ | | | | 68.0 | 78.8 | 94.2 | 73.8 | 83.8 | 95.9 | 39.3 | 79.9 | 73.0 |
| | ✓ | ✓ | ✓ | | | 68.1 | 78.7 | 93.9 | 73.1 | 83.4 | 95.1 | 39.9 | 80.2 | 72.3 |
| | ✓ | ✓ | | ✓ | | 68.5 | 80.1 | **94.2** | 74.2 | 85.1 | 96.0 | 41.7 | 78.9 | 74.3 |
| | ✓ | ✓ | | ✓ | ✓ | **70.1** | **82.2** | 94.1 | **75.2** | **85.8** | **96.3** | **42.2** | **81.7** | **74.5** |

gradually incorporate pseudo-labels without introducing excessive noise. Note that CDSS-Det corresponds to the last row in the table.

To investigate the impact of curriculum learning, we compare CDSS-Det using fixed pseudo-labeling loss coefficients against our dynamic curriculum strategy. As shown in Figure 4, using fixed coefficients leads to unstable performance, with mAP values fluctuating between 74.7 and 75.6 in WORD dataset and fluctuating between 66.8 and 68.5 in TotalSegmentator dataset depending on the coefficient. Notably, large coefficients such as 1.0 and 2.0 result in degraded performance in both two datasets due to training divergence. In contrast, our curriculum-based dynamic weighting strategy achieves a significantly higher mAP of 76.6 in WORD dataset and 70.1 in TotalSegmentator dataset, demonstrating its ability to balance supervision and mitigate overfitting or label noise during training. This highlights that dynamic pseudo loss weighting is crucial for stable and effective semi-supervised learning in medical detection scenarios.

To investigate the effect of weak-strong augmentation in the context of medical image detection, we compare CDSS-Det's performance with and without this strategy. As shown in Figure 4, applying weak-strong augmentation decreases the mAP from 76.6 to 75.0 in WORD dataset and decreases from 70.1 to 68.3 in TotalSegmentator dataset, contrary to trends observed in natural image domains. For instance, in Adaptive Teacher (Li et al., 2022), weak-strong augmentation significantly boosts detection performance across various cross-domain scenarios. However, in our experiments in 3D CT data, this technique appears to hinder performance. One possible explanation is that aggressive augmentation may distort subtle anatomical structures and degrade the quality of pseudo labels, especially when the model already generates high-quality predictions under weak augmentation alone. This highlights a critical difference between medical and natural image domains, where preserving spatial fidelity is often more important than encouraging invariance through strong perturbations.

Overall, our results demonstrate that CDSS-Det consistently outperforms a range of strategies, from models trained solely on labeled target data and those leveraging pre-

training, to fully supervised approaches. By employing curriculum-guided pseudo-labeling and reliability-aware supervision, CDSS-Det effectively mitigates domain shifts and improves cross-domain generalization.

## 4. Conclusion

We demonstrated that CDSS-Det effectively leverages labeled and unlabeled data for cross-domain 3D organ detection, surpassing both pre-trained and fully supervised models. Pseudo-label refinement contributes the most to performance gains, while replay and domain adaptation further enhance generalization. These findings highlight the potential of semi-supervised learning not only to reduce annotation efforts and enhance detection robustness but also to address the domain shifts inherent in medical imaging. By facilitating reliable domain transfer, our approach takes a crucial step toward translating organ detection approaches into clinical practice. Future work will explore extending this framework to additional imaging modalities and anatomical structures to further validate its effectiveness.

## 5. Acknowledgments

The authors gratefully acknowledge the computational and data resources provided by the Leibniz Supercomputing Centre (https://www.lrz.de).

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
