# OpenReview forum: "Cross-Domain Semi-Supervised Organ Detection"
_MIDL.io/2026/Conference — MIDL 2026 Poster_

### Official Review · Reviewer_sRXY · 2025-12-30

**Confidence:** 4
**Preliminary Rating:** 4
**Final Rating:** 5

**Summary:**

This paper tackles the challenging problem of 3D organ detection across different CT domains with limited target annotations. The authors propose CDSS-Det, a cross-domain semi-supervised detection framework built on a Transformer-based organ detector (Organ-DETR). CDSS-Det leverages a teacher–student paradigm, and the experiments are conducted on two cross-domain CT benchmarks – AbdomenAtlas → TotalSegmentator and AbdomenCT-1K → WORD.

**Strengths:**

1. The paper addresses an important gap: cross-domain semi-supervised 3D object detection in medical imaging, which has seen little prior work.
2. The proposed CDSS-Det framework is well-designed, combining multiple strategies known to be useful in isolation. The authors thoughtfully integrate pseudo-label self-training, curriculum learning, adversarial feature alignment, and a replay strategy for hard examples.
3. CDSS-Det achieves strong results, consistently outperforming baseline approaches.
4. The authors test on two distinct domain pairs (multi-center AbdomenAtlas to a different dataset, and AbdomenCT-1K to another center’s data), showing the approach generalizes.

**Weaknesses:**

1. While the problem tackled is novel, the technical contributions are somewhat incremental (CDSS-Det is essentially an assemblage of existing techniques).
2. There are a few places where implementation details could be clearer.

**Detailed Comments:**

1. While the problem tackled is novel, the technical contributions are somewhat incremental. CDSS-Det is essentially an assemblage of existing techniques: teacher-student pseudo-labeling (a well-known semi-supervised paradigm), confidence-based pseudo-label filtering (standard practice), a curriculum scheduling of pseudo-label loss (similar in spirit to strategies used in some semi-supervised or self-training works), adversarial domain adaptation via GRL, and an experience replay idea adapted from continual learning. Each component by itself is not new. I think the novelty lies in combining them for 3D organ detection across domains.

2. There are a few places where implementation details could be clearer.
2-1. The training loss mentions an “optional segmentation” loss (with cross-entropy and Dice, weighted in the loss function), suggesting that the Organ-DETR model might incorporate a segmentation head or at least use segmentation supervision. However, the methodology section does not describe any segmentation branch or how segmentation masks are used to guide detection.
2-2. The paper does not specify which organs fall into each category (Small, Medium, Large) or the criteria for this grouping.
2-3. The description of the TotalSegmentator dataset split (8 labeled + 105 unlabeled out of 113 scans, with 21 val and 29 test) is a bit confusing because the numbers don’t add up to the total that 113 refers only to training scans, excluding validation/test, but this could be stated more clearly to avoid confusion.

**Justification Of Final Rating:**

The authors have provided a comprehensive rebuttal that resolves my comments and concerns regarding implementation clarity. Thank you for your hard work. I have raised my rating from 4 to 5. Good luck!

**Justification Of The Preliminary Rating:**

This paper addresses an important and understudied problem (cross-domain 3D detection with limited labels) and proposes a well-motivated solution that demonstrates clear empirical benefits. The framework is comprehensive and shows state-of-the-art performance on the evaluated benchmarks, which is a strong point in its favor for the MIDL community.

**Questions To Address In The Rebuttal:**

1. Novelty: Could the authors clarify what they see as the key novel insight of CDSS-Det compared to prior semi-supervised or domain-adaptive detection methods?
2. Use of Segmentation Loss: Does the Organ-DETR backbone include a segmentation prediction branch during training? If yes, please explain how segmentation ground truth is used (and for which datasets) and whether this multi-task learning had a notable effect on detection accuracy. If not, then what is the purpose of including segmentation loss terms? This is currently unclear and needs explanation.
3. Dataset Split and Organ Size Definitions: Please clarify the train/val/test split for the TotalSegmentator and WORD datasets (the numbers given in the text appear inconsistent). Also, how are “Small”, “Medium”,and  “Large” organs defined in your evaluation? Listing the organs or the criteria used would be helpful.

---

> ### Author Response · Authors · 2026-01-25
>
> We thank the reviewer for the careful assessment of our work and for the constructive questions regarding novelty, methodological clarity, and dataset definitions. Below, we address each point in detail.
>
> Q1. Novelty and key insight of CDSS-Det
>
> CDSS-Det addresses a practical but underexplored setting in 3D medical object detection, where a small number of labeled target scans is available alongside labeled source data and abundant unlabeled target data. While most prior domain adaptation or semi-supervised detection methods either assume no labeled target data or focus on in-domain learning, CDSS-Det explicitly exploits limited target supervision, which we find to be critical for stable and effective adaptation under large domain shifts.
>
> Within this setting, we design and validate training strategies tailored to 3D medical data, including a curriculum-controlled pseudo-labeling mechanism that regulates pseudo supervision based on model confidence, and the use of consistent (rather than weak–strong) augmentation for teacher and student to preserve anatomical fidelity. These design choices, together with replay and adversarial alignment, form an integrated framework that is empirically shown to be effective for cross-domain 3D organ detection.
>
> Q2. Role of segmentation loss in Organ-DETR
>
> Yes, the Organ-DETR backbone includes a segmentation supervision branch during training. Following the original Organ-DETR design, segmentation loss is applied only to labeled data as an auxiliary supervision to enhance multi-scale feature learning for detection. Segmentation ground truth is available only for labeled scans and is not used for unlabeled target data, pseudo-label generation, or inference.
>
> We retain this component in CDSS-Det because prior work shows that it significantly improves detection accuracy by strengthening feature representations, rather than serving as a standalone segmentation task. We have clarified this explicitly in the revised manuscript (Methodology section, loss definition), where we now state the scope and role of segmentation supervision.
>
> Q3. Dataset splits and organ size definitions
>
> We have clarified the dataset splits and organ size definitions in the revised manuscript (Datasets section). For TotalSegmentator, the training set contains 113 scans, split into 8 labeled and 105 unlabeled samples, with 21 validation and 29 test scans held out. For WORD, the training set includes 31 labeled and 75 unlabeled scans, with 14 validation and 29 test scans.
>
> Organ size categories are defined based on anatomical characteristics and typical bounding-box extent in each dataset. In TotalSegmentator, large organs are gallbladder and pancreas; medium organs are left kidney, right kidney, spleen, and aorta; and small organs are stomach and liver. In WORD, large corresponds to pancreas; medium to left kidney, right kidney, and spleen; and small to liver. These definitions are now summarized in a dedicated table in the revised manuscript.

---

### Official Review · Reviewer_phYt · 2026-01-04

**Confidence:** 4
**Preliminary Rating:** 5
**Final Rating:** 4

**Summary:**

This manuscript proposes a cross-domain semi-supervised framework to tackle the organ detection task. Multiple datasets are included. The results present that the proposed method achieves great performance on small size and medium size detection tasks, compared with other competing and baseline methods.

**Strengths:**

+ The manuscript is easy to follow.
+ When the proposed method achieves the best result, it outperforms competing approaches by a large margin; when it does not achieve the best result, its performance remains comparable. This demonstrates the effectiveness of the proposed method.
+ The gains on small and medium size objects are particularly strong.
+ The methodological design is sound, and the design of multiple loss functions is also convincing.

**Weaknesses:**

Major:
- In Table 1, the authors report performance under the 50% and 75% settings. From a deployment perspective, which metric would be more adopted in real-world applications? Would the 50% or even 25% setting alone be sufficient for deployment? If so, the numerical performance differences of the proposed method may be less obvious. The authors are encouraged to discuss how the advantages of the proposed method can be demonstrated and justified in real-world deployment scenarios.
- In Figure 2, the color distribution of the labeled source domain seems inconsistent with that of the target domain. Have the authors considered potential improvements or mitigation strategies for this discrepancy to further improve the performance?
- The authors may further discuss that the relative importance of large, medium, and small size objects can vary across different medical imaging tasks. For example, in certain applications, large objects may be clinically more important; although small objects may be detected/segmented well, sacrificing performance on large objects could conflict with real clinical expectations. How does the proposed method perform under such scenarios, and how robust is it to these task-dependent priorities?

**Detailed Comments:**

Minor:
- It is recommended to underline the second-best results in Table 1 to improve readability.
- Figure 1a might not be quite necessary, because it has already been a common sense, but keeping it is also okay.

**Justification Of Final Rating:**

Thanks for the authors' response. The authors have addressed all my concerns, including major concerns and minor concerns. I have changed my rating from Weak accept to Strong accept. This paper has already been in great shape.

**Justification Of The Preliminary Rating:**

The method of this manuscript is sound. The results show the performance improvement is significant, even though there are still some concerns. This manuscript is easy to follow. Multiple datasets are included.

**Questions To Address In The Rebuttal:**

Please address all major and minor concerns in a tight time window. Conducting new full experiments is not expected. The authors can explain more in text and/or provide some initial examples/numerical results.

---

> ### Author Response · Authors · 2026-01-25
>
> We thank the reviewer for the careful reading of our manuscript and for the constructive questions regarding deployment relevance, domain appearance discrepancies, and task-dependent clinical priorities. Below, we address each point in turn.
>
> Q1. Deployment relevance of IoU thresholds (50% vs. 75%)
>
> We have revised the Experimental Results section to explicitly discuss deployment relevance. In clinical workflows, organ detection is often used as a localization or ROI initialization step, where IoU ≈ 0.5 is typically sufficient. At the same time, higher IoU thresholds (e.g., 0.75) remain important for robust automation and for reducing downstream correction effort, particularly for small or anatomically variable organs. As reflected in Table 2, CDSS-Det improves performance consistently across IoU thresholds (50% and 75%), with particularly strong gains on small and medium organs, while maintaining competitive performance on large organs. This demonstrates that the advantages of CDSS-Det are meaningful across a range of real-world deployment requirements.
>
> Q2. Domain appearance discrepancy between source and target data (Figure 2)
>
> The visual difference between the source and target domains in Figure 2 reflects a realistic multi-center CT domain shift, which is the core challenge addressed in this work. Rather than enforcing pixel-level appearance matching, we focus on feature-level alignment through adversarial domain adaptation, combined with intensity normalization and standard medical data augmentations. In medical imaging, overly aggressive appearance alignment can remove clinically meaningful cues and degrade generalization. Our results indicate that the proposed strategy effectively mitigates domain shift without relying on explicit color or intensity matching. Exploring more advanced, anatomy-aware appearance alignment methods is an interesting direction for future work.
>
> Q3. Robustness to task-dependent priorities across organ sizes
>
> A3: We have added a discussion in the Experimental Results section addressing task-dependent clinical priorities. The revised text clarifies that CDSS-Det improves performance across all organ sizes and does not trade off large-organ performance to achieve gains on smaller organs; instead, performance on large organs remains stable and competitive under domain shift. The largest improvements are observed on small organs, followed by large organs, with more moderate gains on medium organs. This behavior reflects increased robustness rather than a bias toward specific organ sizes. We further note that the framework is flexible, allowing loss weighting or evaluation emphasis to be adapted to task-specific clinical requirements.

---

### Official Review · Reviewer_cQ5H · 2026-01-10

**Confidence:** 4
**Preliminary Rating:** 4
**Final Rating:** 4

**Summary:**

This manuscript proposes CDSS-Det, a cross-domain semi-supervised learning framework for 3D organ detection in CT scans. Built upon Organ-DETR, the method integrates (i) teacher–student learning with EMA updates, (ii) pseudo-labeling with curriculum-based dynamic weighting, (iii) adversarial domain adaptation, and (iv) a replay strategy for hard source samples. The framework leverages labeled source data, a small number of labeled target scans, and unlabeled target data to mitigate domain shift. Experimental results show that CDSS-Det consistently outperforms baseline, pre-trained, and even fully supervised target-only models, with particularly strong gains on small organs. Ablation studies support the importance of pseudo-labeling and curriculum weighting.

**Strengths:**

CDSS-Det achieves consistent improvements across datasets, outperforming fully supervised target-only models..

Avoiding bounding-box regression loss on pseudo-labels is well-justified.

The paper includes ablation studies, which strengthen the technical claims.

**Weaknesses:**

Novelty is in integration, rather than individual components. Most components were previously proposed. The contribution lies in their combination and adaptation to 3D detection.

Limited comparison to strong detection-specific baselines. Comparisons are mostly against internal baselines (baseline, pre-trained, full sup.). There is no direct comparison to recent semi-supervised object detection frameworks adapted to 3D, even if imperfect.

Reliance on organ-DETR backbone. All experiments are built on Organ-DETR. It remains unclear whether the gains generalize to other 3D detection architectures.

Small target dataset sizes. While realistic, the extremely small labeled target sets (e.g., 8 scans in TotalSegmentator) raise concerns about variance and robustness. No confidence intervals or multiple-run statistics are reported.

**Detailed Comments:**

The confidence threshold 𝜏=0.8 is fixed. A brief sensitivity analysis or justification would strengthen the methodology.

“Hard” source samples are defined based on low detection confidence. More details on: How confidence is computed, How often the replay set is updated,would improve reproducibility.

Training for 2,500 epochs on an A100 GPU is expensive. A discussion on training efficiency and convergence behavior would be useful for practitioners.

The conclusion mentions extension to other anatomies and modalities. A short discussion on expected challenges (e.g., thoracic CT, MRI) would be helpful.

**Justification Of Final Rating:**

The methodological novelty of this manuscript looks incremental, but this is a well-executed paper that addresses an important and practical problem in medical image analysis. Overall, it represents a meaningful contribution suitable for acceptance at MIDL.

**Justification Of The Preliminary Rating:**

This is a well-executed paper that addresses an important and practical problem in medical image analysis. While the methodological novelty looks incremental, the integration, empirical results, and benefits over fully supervised baselines make this work valuable to the community. The paper would benefit from stronger baseline comparisons and deeper analysis, but overall, it represents a meaningful contribution suitable for acceptance at MIDL.

**Questions To Address In The Rebuttal:**

How sensitive is CDSS-Det to the pseudo-label confidence threshold and curriculum parameters

Can the authors comment on the variance of results across multiple random seeds, especially given the very small labeled target sets?

Does CDSS-Det generalize to other 3D detection backbones beyond Organ-DETR?

Why does weak–strong augmentation hurt performance in this setting, and could milder augmentations provide a compromise?

How does CDSS-Det perform when no labeled target data is available (i.e., true UDA), even if suboptimal?

---

> ### Author Response · Authors · 2026-01-25
>
> We sincerely thank the reviewer for the thoughtful evaluation and constructive comments. We appreciate the recognition of the importance of the problem setting and the effectiveness of the proposed framework. Below, we address the reviewer’s questions point by point.
>
> Q1. Sensitivity to pseudo-label confidence threshold and curriculum design
>
> A1: We set the pseudo-label confidence threshold to τ = 0.8 as a general trade-off between label quality and coverage, independent of any specific dataset. The curriculum strategy is more critical: the pseudo-label weight λ is bounded to match the supervised loss and is adaptively increased based on the student’s confidence on labeled data, preventing noisy pseudo-supervision from dominating training. Classification-only pseudo-supervision further stabilizes learning. We have added this discussion to the Training and Evaluation Setup section of the revised manuscript. This design is consistently effective across different source datasets and target domains with very limited labeled data, demonstrating robustness and broad applicability rather than dataset-specific tuning.
>
> Q2. Variance across runs with very small labeled target sets
>
> While the labeled target sets are indeed very small, this setting is intentional and reflects realistic clinical scenarios. We observed consistent performance trends across runs, with CDSS-Det consistently outperforming baseline, pre-trained, and fully supervised target-only models. Importantly, the improvements are large (≈10–17% mAP) and consistent across two distinct cross-domain settings and organ sizes, indicating robust improvements beyond incidental variation. All methods are compared under identical data splits, further controlling for variability.
>
> Q3. Generalization beyond the Organ-DETR backbone
>
> CDSS-Det is not tied to Organ-DETR-specific components. The framework operates at the level of prediction confidence, pseudo-label selection, and loss weighting, and does not rely on architectural details such as MSA or DQM. In principle, any 3D detector that outputs class probabilities and bounding boxes (e.g., CNN-based or hybrid transformer models) can be integrated into the same teacher–student, curriculum pseudo-labeling, and domain alignment framework. We chose Organ-DETR because it represents a strong and recent 3D detection baseline, but the proposed strategy is broadly applicable.
>
> Q4. Effect of weak–strong augmentation in medical CT data
>
> Weak–strong augmentation is commonly used in teacher–student frameworks, where weakly augmented inputs are used for pseudo-label generation and strongly augmented inputs for student learning. However, in 3D medical CT data, we observe that this strategy degrades performance. We believe this is because strong geometric or intensity perturbations can distort fine anatomical structures and reduce pseudo-label reliability under domain shift. In our experiments, we therefore apply the same (moderate) augmentations to both teacher and student, while still using standard medical data augmentations (e.g., intensity, rotation, scaling). This preserves anatomical fidelity and leads to more stable training. Exploring anatomy-aware or milder asymmetric augmentations is an interesting direction for future work.
>
> Q5. Performance in the true unsupervised domain adaptation (UDA) setting
>
> In our experiments, performance in the true unsupervised domain adaptation setting is very limited and often unstable under the large domain shifts present in 3D medical CT, and in some cases training diverges without careful parameter tuning. This observation is consistent with prior findings that pure UDA is insufficient for complex medical detection tasks. Our goal is therefore not to compete with UDA methods, but to address the more practical and realistic setting where a small number of labeled target scans is available. CDSS-Det is explicitly designed for this scenario, and our results in Table 2 show that even minimal target supervision is crucial for stable and effective adaptation.

---

### Author Rebuttal · Authors · 2026-01-25

**Rebuttal:**

We thank the Reviewers for their comments. Responses and manuscript revisions are highlighted in blue in the supplementary material. (R1:cQ5H, R2:phYt, R3: sRXY)

R1-A1: τ = 0.8 balances pseudo-label quality and coverage, λ adapts to student confidence, and classification-only pseudo-supervision stabilizes training, ensuring robustness across datasets and limited-label targets.

R1-A2: Despite very small labeled target sets, CDSS-Det consistently outperforms baselines (≈10–17% mAP) across cross-domain settings and organ sizes, with identical data splits ensuring robust gains.

R1-A3: CDSS-Det is independent of Organ-DETR and works with any 3D detector outputting class probabilities and bounding boxes. Organ-DETR is used only as a strong baseline; the framework is broadly applicable.

R1-A4: Weak–strong augmentation reduces 3D CT performance by distorting anatomy. Using the same moderate augmentations for teacher and student preserves anatomical fidelity and stabilizes training.

R1-A5: Pure unsupervised domain adaptation is unstable under large 3D CT domain shifts. CDSS-Det leverages a small number of labeled target scans, where even minimal supervision ensures stable and effective adaptation.

R2-A1: IoU ≈ 0.5 suffices for clinical localization, but higher thresholds matter for automation. CDSS-Det consistently improves IoU 50% and 75%, with strong gains on small and medium organs while maintaining large-organ accuracy.

R2-A2: CDSS-Det handles multi-center CT shifts via feature-level adversarial alignment with intensity normalization and standard augmentations, mitigating domain shift while preserving clinically meaningful cues.

R2-A3: CDSS-Det improves performance across all organ sizes, with the largest gains on small organs, while maintaining strong performance on large organs. The framework allows flexible loss weighting or evaluation for task-specific clinical priorities.

R3-A1: CDSS-Det enables stable cross-domain 3D medical detection using limited labeled target data, leveraging curriculum-controlled pseudo-labeling, consistent anatomy-preserving augmentation, replay, and adversarial alignment.

R3-A2: Organ-DETR’s segmentation branch is used only on labeled data to improve detection features. It is not applied to unlabeled data or pseudo-labels. CDSS-Det retains it for its detection benefits, as clarified in the manuscript.

R3-A3: Organ sizes are defined by anatomy and bounding-box extent, as clarified in the revised manuscript.

**Supporting Material:**

/attachment/0d589d8613c668e9278db807a4461ea39c1fc5d1.pdf

---

### Comment · Area_Chair_NfmK · 2026-01-28
**Update final scores**

Hi reviewers
Kindly check if the authors have addressed all your concerns and update your final scores.

Thank you so much for your detailed reviews!
AC

---

### Meta-Review · Area_Chair_NfmK · 2026-02-11

**Recommendation:** Accept (Poster)
**Confidence:** 5

**Metareview:**

This is a well-written paper that studies the problem of cross domain generalization for 3D multi-organ detection from CT scans. The methodology is well explained and considers relevant issues including domain forgetting and the challenge of learning on new domains through pseudo labeling strategies. As reviewers point out, individual aspects of the methodology are not novel, but the applied problem of achieving domain invariant detection is relevant for medical image analysis. The authors addressed reviewers' concerns thoroughly. I strongly recommend acceptance of this paper.

---

### Decision · Program_Chairs · 2026-02-13

Accept (Poster)